# *"It might be a statistic to me, but every death matters."*: An assessment of facility-level maternal and perinatal death surveillance and response systems in four sub-Saharan African countries

Mary V. Kinney[1,2]*, Gbaike Ajayi[3,4], Joseph de Graft-Johnson[1,3], Kathleen Hill[3,4], Neena Khadka[1,3], Alyssa Om'Iniabohs[1,3], Fadzai Mukora-Mutseyekwa[5], Edwin Tayebwa[6], Oladapo Shittu[7], Chrisostom Lipingu[8], Kate Kerber[1], Juma Daimon Nyakina[8], Perpetus Chudi Ibekwe[9], Felix Sayinzoga[10], Bernard Madzima[11], Asha S. George[2], Kusum Thapa[3,4]

**1** Save the Children US, Washington, DC, United States of America, **2** University of the Western Cape, Cape Town, South Africa, **3** US Agency for International Development (USAID)'s Maternal and Child Survival Program (MCSP), Washington, DC, United States of America, **4** Jhpiego, Baltimore, Maryland, United States of America, **5** USAID's Maternal and Child Health Integrated Program/John Snow Inc., Harare, Zimbabwe, **6** USAID's MCSP/Jhpiego, Kigali, Rwanda, **7** Ahmadu Bello University, Zaria, Kaduna State, Nigeria, **8** Bukoba Regional Referral Hospital, Kagera, Tanzania, **9** Maternal and perinatal death surveillance and response, Abakaliki, Ebonyi State, Nigeria, **10** Maternal, Child, and Community Health Division, Rwanda Biomedical Center, Kigali, Rwanda, **11** Family Health Directorate, Ministry of Health and Child Care, Harare, Zimbabwe

* mkinney@uwc.ac.za

## Abstract

### Background

Maternal and perinatal death surveillance and response (MPDSR) systems aim to understand and address key contributors to maternal and perinatal deaths to prevent future deaths. From 2016–2017, the US Agency for International Development's Maternal and Child Survival Program conducted an assessment of MPDSR implementation in Nigeria, Rwanda, Tanzania, and Zimbabwe.

### Methods

A cross-sectional, mixed-methods research design was used to assess MPDSR implementation. The study included a desk review, policy mapping, semistructured interviews with 41 subnational stakeholders, observations, and interviews with key informants at 55 purposefully selected facilities. Using a standardised tool with progress markers defined for six stages of implementation, each facility was assigned a score from 0–30. Quantitative and qualitative data were analysed from the 47 facilities with a score above 10 ('evidence of MPDSR practice').

### Results

The mean calculated MPDSR implementation progress score across 47 facilities was 18.98 out of 30 (range: 11.75–27.38). The team observed variation across the national MPDSR

**Data Availability Statement:** All relevant data are within the manuscript and its Supporting information files.

**Funding:** This study was made possible by the generous support of the American people through the US Agency for International Development (USAID) under the terms of the Cooperative Agreement AID-OAA-A-14-00028 (www.usaid.gov). The contents are the responsibility of the authors and do not necessarily reflect the views of USAID or the United States Government. AG is supported by the South African Research Chair's Initiative of the Department of Science and Technology and National Research Foundation of South Africa (Grant No. 82769) (www.nrf.ac.za/tags/dst), and the South African Medical Research Council (www.mrc.ac.za). Any opinion, finding, and conclusion or recommendation expressed in this material is that of the author, and the National Research Foundation of South Africa does not accept any liability in this regard.

**Competing interests:** The authors declare that they have no competing interests.

guidelines and tools, and inconsistent implementation of MPDSR at subnational and facility levels. Nearly all facilities had a designated MPDSR coordinator, but varied in their availability and use of standardised forms and the frequency of mortality audit meetings. Few facilities (9%) had mechanisms in place to promote a no-blame environment. Some facilities (44%) could demonstrate evidence that a change occurred due to MPDSR. Factors enabling implementation included clear support from leadership, commitment from staff, and regular occurrence of meetings. Barriers included lack of health worker capacity, limited staff time, and limited staff motivation.

## Conclusion

This study was the first to apply a standardised scoring methodology to assess subnational- and facility-level MPDSR implementation progress. Structures and processes for implementing MPDSR existed in all four countries. Many implementation gaps were identified that can inform priorities and future research for strengthening MPDSR in low-capacity settings.

## Introduction

Despite gradual progress, women and their babies continue to die of complications of gravidity and childbirth or complications in the first month after birth; an estimated 303,000 global maternal deaths, 2.6 million stillbirths, and 2.5 million newborn deaths occur per year [1,2]. Over 40% of these deaths occur in sub-Saharan Africa, and one-half occur in the perinatal period [3–5]. Many of these deaths are preventable through timely access to high-quality, safe care that delivers evidence-based interventions and avoids harmful practices for women and newborns during gravidity, childbirth, and the postnatal period [5]. To achieve the Sustainable Development Goal targets to end preventable maternal and newborn deaths by 2030, there has been a renewed focus on improving quality of care [6,7], as reflected in multiple global and country efforts [8–14]. Concurrently, there has been momentum to strengthen maternal and perinatal death surveillance and response (MPDSR) as one mechanism to help address quality of care deficits and other important contributors to preventable maternal and newborn deaths [15–18].

MPDSR is a systematic process used to understand the medical causes and the modifiable factors that contribute to maternal and perinatal deaths to identify actions to prevent future deaths [18]. MPDSR operates at all levels of the health system. Its aims are to ensure accurate documentation and reporting of deaths, identify modifiable systemic and social factors at various levels (e.g., delays in care seeking, lack of access to care, quality of care gaps), and link recommendations and accountability for follow-up actions [19–21].

The World Health Organization (WHO) has distinct guidelines for maternal death surveillance and response and for perinatal death audit [19,20]. WHO promotes an integrated approach when appropriate, and many countries have adopted integrated national MPDSR guidelines and policies in recent years [17,21]. A number of studies and reviews have explored facilitators and inhibitors of implementation or sustainability of maternal and perinatal mortality audit systems [17,21–24]. Challenges to effective implementation of MPDSR have been identified, including not having a national MPDSR policy, weak information and surveillance systems (e.g., lack of vital registration systems and lack of primary data on cause of death), lack of diagnostic capacity for accurate classification of cause of death, and gaps in identifying and

documenting maternal and perinatal deaths. Even when data do exist and deaths are reviewed, identified modifiable factors may not be addressed, undermining the "response" component of MPDSR [21,25].

Despite some knowledge of the high-level factors enabling or preventing implementation, there is limited understanding of subnational and facility-based MPDSR activities in sub-Saharan African countries. Better understanding of MPDSR implementation status at subnational and facility levels, including enablers and barriers, can help countries to strengthen MPDSR systems as an important element of their efforts to reduce preventable deaths.

## Methodology

### Aim and design

The aim of this study was to systematically assess the level of implementation of MPDSR in four sub-Saharan African countries, applying a standardised scoring methodology, and to describe common facilitators and barriers to sustainable MPDSR practice. A cross-sectional, mixed-methods research design was used to assess MPDSR implementation at subnational and facility levels. Qualitative and quantitative data collection methods were employed, including observations (e.g., onsite review of facility documents) and semistructured key informant interviews with subnational and facility managers and staff. The US Agency for International Development (USAID)'s Maternal and Child Survival Program (MCSP) led the study with support from ministries of health. Country visits took place between October 2016 and May 2017. Country study protocols and tools were approved by in-country ethics committees, including the Rwanda National Ethics Committee, Tanzania's National Institute for Medical Research, the Medical Research Council of Zimbabwe, and Nigeria's National Health Research Ethics Committee. The study received a nonhuman subjects research determination by the Johns Hopkins Bloomberg School of Public Health Institutional Review Board. The data collected in this assessment did not include any personal identifiers from respondents. Before review of facility documents and before every key informant interview, the interviewer read an oral consent script and asked the participant to respond "yes" or "no". Oral consent was obtained in Nigeria, Rwanda, and Zimbabwe and written consent obtained in Tanzania, in accordance with ethics committee approvals in each local setting.

### Sampling

Four countries—Nigeria, Rwanda, Tanzania, and Zimbabwe—were purposively selected as countries from which a more detailed picture of district- and facility-based MPDSR activities could be gathered. Factors that influenced the selection of the four countries included: (1) having existing national guidelines for MPDSR (or any form of maternal and/or perinatal death audit policy), (2) country government interest and approval, (3) in-country presence of MCSP (or affiliated organization) to support the assessment, and (4) presence of other in-country partners supporting maternal and/or perinatal death review and response. Table 1 presents selected statistics for the four countries, demonstrating the range of maternal and perinatal death rates and ratios, and institutional birth coverage across the four countries.

National and subnational stakeholders were identified for interview by MCSP in-country staff and/or the ministry of health. A total of 41 stakeholders were interviewed, including four national stakeholders in Zimbabwe and Tanzania, and 37 regional and district government health officials supporting MPDSR in Zimbabwe, Tanzania, and Nigeria. No stakeholder interviews were conducted in Rwanda due to the unavailability of identified interviewees, who were all engaged in a national meeting at the time of the assessment. Selection of facilities was purposeful and done in collaboration with the ministries of health and included the following

**Table 1. Selection of maternal and newborn health information for the four countries.**

| Indicator | Nigeria | Rwanda | Tanzania | Zimbabwe |
|---|---|---|---|---|
| Total live births (2015) | 7132700 | 362600 | 2064400 | 538600 |
| Maternal mortality ratio, deaths per 100,000 live births (2015) | 814 | 290 | 398 | 443 |
| Neonatal mortality rate, deaths per 1,000 live births (2015) | 34 | 17 | 22 | 23 |
| Stillbirth rate per 1,000 total births (2015) | 42.9 | 17.3 | 22.4 | 20.6 |
| Institutional delivery (2010–2015) | 36% | 91% | 80% | 50% |
| Total fertility rate (2015) | 5.6 | 3.8 | 5.1 | 3.9 |
| History of MPDSR | Different pilot programmes initiated before 2016; national MPDSR guidelines adopted in 2015. | Maternal mortality audits started at some hospitals in 2009; neonatal audits started in 2010, and stillbirth audits started in 2015. | Some facilities have a long history of maternal death audits. Wide-scale maternal and perinatal death audits started in 2006; national MPDSR guidelines adopted in 2015. | Maternal and perinatal death audits started in central hospitals 30 years ago; national MPDSR guidelines adopted in 2013. |

Source: Data extracted from Healthy Newborn Network [26].

criteria: provision of childbirth services and current or previous experience conducting maternal and/or perinatal death audits. Facilities were based on a convenience sample rather than a true probability sample and differed between countries with respect to geographic spread and levels of care. For example, two regions (states) were targeted in Nigeria and Tanzania due to MCSP presence in these areas at the time of the assessment, whereas facilities in all major geographic areas were targeted in Rwanda and Zimbabwe. In total, 55 health facilities (41 hospitals and 14 health centres) received onsite visits. Table 2 summarises the geographic distribution and types of facilities and subnational stakeholders selected in each country.

**Table 2. Summary of facility and stakeholder samples.**

| | Nigeria | Rwanda | Tanzania | Zimbabwe | TOTAL |
|---|---|---|---|---|---|
| **Total Number of Facilities Assessed** | 10 | 13 | 26 | 16 | 55 |
| **Facility Type** | | | | | |
| **Number of health centres** | 4 | 3 | 7 | 0 | 14 |
| **Number of hospitals** | 6 | 10 | 9 | 16 | 41 |
| **Total Number of Stakeholders Interviewed*** | 7 | 0 | 17 | 17 | 41 |
| **Stakeholder Type** | | | | | |
| National | 0 | 0 | 1 | 3 | 4 |
| Subnational province/state/region | 2 | 0 | 2 | 5 | 9 |
| Subnational district/local government area | 4 | 0 | 14 | 8 | 26 |
| Other | 1 | 0 | 0 | 1 | 2 |
| **Geography Covered** | 2 states | national | 2 regions | national | |
| Estimated population in 2016 | Ebonyi: 2880000 Kogi: 4473000 | National: 11669000 | Kagara: 2790000 Mara: 1924000 | National: 14030000 | |

*Key informant stakeholders were primarily subnational (regional/district) government health officials involved with supporting MPDSR at subnational level.

Population data sources: The World Bank Group, Tanzania National Statistics Bureau, Nigeria National Statistics Bureau [27–29].

## Data collection

Data collectors included MCSP technical staff and in-country staff from MCSP partner organisations (Save the Children and Jhpiego), national and subnational ministry of health representatives, professional association members (in Nigeria only), and local consultants as needed. The size of the assessment teams for each facility varied from two to five people. Each country's data collection team received standardised training on completion of the data collection tools and assessment methodology. Data collection tools included a semistructured questionnaire for subnational managers and stakeholders (S1 Table) to explore district and regional MPDSR activities, and subnational support of facility-level MPDSR implementation. The facility assessments included two types of data collection: 1) administration of a standardised, semistructured questionnaire to facility health workers supporting MPDSR-related activities who were present on the day of the visit, and 2) observations by assessors of MPDSR-related documents and activities in the facility (e.g., review of MPDSR meeting notes). Generally, facility-level interviews were conducted with health workers as a team, with individual staff selected by the facility manager.

An implementation tool was developed specifically for this study, adapted from the work by Bergh and colleagues for understanding facility-based kangaroo mother care implementation status [30,31]. The tool designed for this study was developed by grounding the constructs in the literature on the topic, engaging experts in the development of the criteria and consulting global guidelines (Table 3). It was also informed by a set of potential questions and progress markers proposed for measuring the status of perinatal death audit implementation [24].

## Data analysis

To understand the context and history of implementation, a desk review of related national MPDSR guidelines and literature on implementation of MPDSR in these countries was conducted. A linked policy mapping set out to determine the content of each national guideline in relation to instructions that have been provided to subnational and facility levels regarding implementation.

To derive a cumulative implementation progress score for each facility, the quantitative data were analysed using the adapted implementation progress monitoring model. An implementation progress score was calculated for each facility across six stages of implementation, with each stage having a weighted score based on specific points (Fig 1). For each stage, the assessors considered all relevant collected data to assign stage-specific points, contributing to a possible total score of 30 (see Table 3). Any discrepancies between the data collectors' score assignment and progress marker results were resolved through discussion and consensus, with the final score determined by the lead investigators (KK for Zimbabwe, KK and OS for Nigeria, KT and GA for Rwanda, and KT and MK for Tanzania). The lead investigators also met with in-country ministry of health and partner stakeholders before and after assessments to present the study design and discuss interpretation of the findings before scores were finalised. Facilities that scored greater than or equal to 10 met at least the fourth stage of 'evidence of practice'. Eight facilities were excluded from the qualitative and quantitative analyses because they did not meet the facility inclusion criteria of 'evidence of practice' (seven in Nigeria and one in Tanzania).

Data from the facility and subnational key informant questionnaires were extracted into a database to tabulate descriptive means and frequencies of explanatory variables and progress markers (S1 Data). Qualitative data were analysed using thematic content analysis. Team members (KT, MK, and JJ) independently coded qualitative responses, consulted, and reached consensus on data interpretation. The team mapped national guidelines and tools using a

**Table 3. Progress markers and rationale for assessing.**

| Stage of implementation | Progress markers and instrument items | Rationale for instrument items based on the literature and global guidelines |
|---|---|---|
| 1. Creating awareness (**2 points**) | Number and type of (senior) managers involved in implementation process (in relation to size of facility)<br>• Special person(s) who take specific effort in promoting death reviews, including management, professionals, driving forces (contact person, meeting coordinator, other champion)<br>• Clear leader(s) are involved in establishing and championing death reviews (past or future). | Successful implementation of MPDSR requires leaders to champion the process and access change agents at other levels to address larger, systemic concerns identified through MPDSR [21–24,32]. |
| 2. Adopting the concept (**2 points**) | Decision to implement MPDSR<br>• Knowledge of the original decision to implement death reviews. If death reviews have not yet been implemented, has a formal decision been made? | A formal decision by facility leadership and subnational actors supports uptake of implementation after the intervention has been introduced and leadership identified [21,33]. |
| | Steering committee<br>• A death review leadership team or steering committee is established. | A steering committee ensures the overall responsibility for operationalising the audit policy, provides technical assistance for the implementation of audit systems, and monitors recommendations and follow-through [19]. Supervision and teamwork within a supportive environment are essential components to setting the foundation for a functioning MPDSR process [21,24]. |
| 3. Taking ownership (**6 points**) | Tools available<br>• A data collection form is available.<br>• Tools include cause of death.<br>• Tools include modifiable factors.<br>• Tools include a place to follow up on actions taken. | National guidelines with clearly defined roles and responsibilities, tools, and familiarity and confidence in the reporting process enable implementation [21–23]. |
| | Meeting process established<br>• Informants' ability to describe or show documentation of meeting process<br>• A staff meeting conduct agreement is available. | Part of taking ownership involves having team members engaged in the process. This can be undermined if staff feel that MPDSR discussions are not protected, confidential spaces. Specific actions can be taken to create no-blame environment, such as having a code of conduct members agree to adhere to during a review [19]. The lack of trust between health professionals and service administrators, issues around the culture of blame and fear of potential legal ramifications, and lack of ownership in a process prevent successful implementation [21,22]. |
| | Resources allocated<br>• Allocations from the hospital budget or support from other partners to establish death reviews | MPDSR requires staff time and skills, meeting space, and stationery [21–23]. Reliance on external funds and/or goodwill of professional organisations to support the process can be an inhibitor of implementation [23]. |
| 4. Evidence of practice (**7 points**) | Evidence of MPDSR meetings<br>• Meeting minutes are available.<br>• Meeting minutes include action items.<br>• Meeting minutes include follow-up from previous meetings.<br>• Meeting notes respect confidentiality of staff and patients. | Documentation of meeting provides evidence that regular meetings take place and enables reflection on the quality of the meetings [21]. |
| | Orientation for new staff<br>• Face-to-face or written orientation on death reviews is available for new staff. | Face-to-face or written orientation of new staff about the death review process supports implementation efforts, since everyone is onboarded to the process [21]. |
| | MPDSR data use<br>• Data trends are displayed or shared. | Data collection and use are foundations of MPDSR. A number of informative quantitative analyses and outcomes can be tallied by the MPDSR committee or designated staff and presented at scheduled review meetings, as well as posted publically within the ward or unit. Looking at data trends over time, such as numbers of admissions, births, and deaths, as well as trends in causes of death and types of modifiable factors are important components of MPDSR tracking. Improved confidence in data capture, use, and reliability enables implementation [21,23,32]. |

(*Continued*)

**Table 3.** (Continued)

| Stage of implementation | Progress markers and instrument items | Rationale for instrument items based on the literature and global guidelines |
|---|---|---|
| 5. Evidence of routine integration (**7 points**) | Further evidence of practice<br>• There is evidence of change based on recommendations that arise from death review findings. | Implementation is encouraged by evidence of the MPDSR process, leading to change or having improved health services as a results of the process [23]. When problems identified during review meetings are not followed up on and addressed, staff are not motivated and/or lose motivation to participate in MPDSR activities [22,34]. |
| | Evidence of routine MPDSR practice<br>• Death review meetings are held at stated interval (e.g., weekly, monthly). | Holding regular meetings is an important element of integrating MPDSR into routine practice. Most national policies stipulate that MPDSR committees meet regularly [21,24]. |
| | Multidisciplinary meetings<br>• Death review meetings include staff from different disciplines and management. | Participation of all health worker cadres involved in the process of caring for women and newborns enhances the analysis of death information and the identification and implementation of follow-up actions to address modifiable factors [19,24]. |
| | Community linkages<br>• There is evidence of reporting findings and progress to the community. | Regular feedback of results to communities and to subnational level ensures accountability and promotes sustainability [21]. Institutionalising MPDSR supported by communities strengthens collective ownership, responsibility, and quality of care [22]. |
| 6. Evidence of sustainable practice (**6 points**) | Documented results<br>• Facility records show ongoing death review meetings for at least 1 year. | Regular audit meetings practised over a long time reflect sustained practice; staff have an expectation that meetings will occur [21,24]. |
| | Evidence of staff development<br>• There is a plan in place to ensure all staff receive MPDSR training.<br>• There is evidence that staff have received MPDSR training in the past year. | Depending on the role and level of implementation of the audit system, district health staff, administrative staff, health workers, and other relevant stakeholders require initial and/or regular training specific to their role in the audit process [19,21,24]. |
| | Score on the first five stages (divided by 12) | Sustainable practice is influenced by the level of implementation of elements in the first five stages. |

*MPDSR = maternal and perinatal death surveillance and response.

content analysis and verified data with national stakeholders (S2 Table). To determine the leading facility-reported barriers and enablers to MPDSR implementation, the team analysed the frequency of qualitative responses from facility interviews based on the thematic content analysis and considered the frequency of relevant progress markers (S3 Table).

## Results

### National and subnational enabling structures

The history of introducing and implementing maternal and perinatal death audits or reviews varied among the four countries (S1 File). National MPDSR guidelines, tools, and forms varied in content across the four countries, including guidance on methods to classify deaths and timeline for death notification (S2 Table). Paper-based systems were used in all four countries. In addition, Rwanda used electronic tools for documenting and reporting maternal deaths, and one province in Zimbabwe was piloting an electronic data system for both maternal and neonatal data. Subnational managers interviewed in Tanzania, Nigeria, and Zimbabwe expressed concerns about the quality of data in facility MPDSR reports in their district or region. All countries had active national MPDSR committees, but subnational support structures varied among countries.

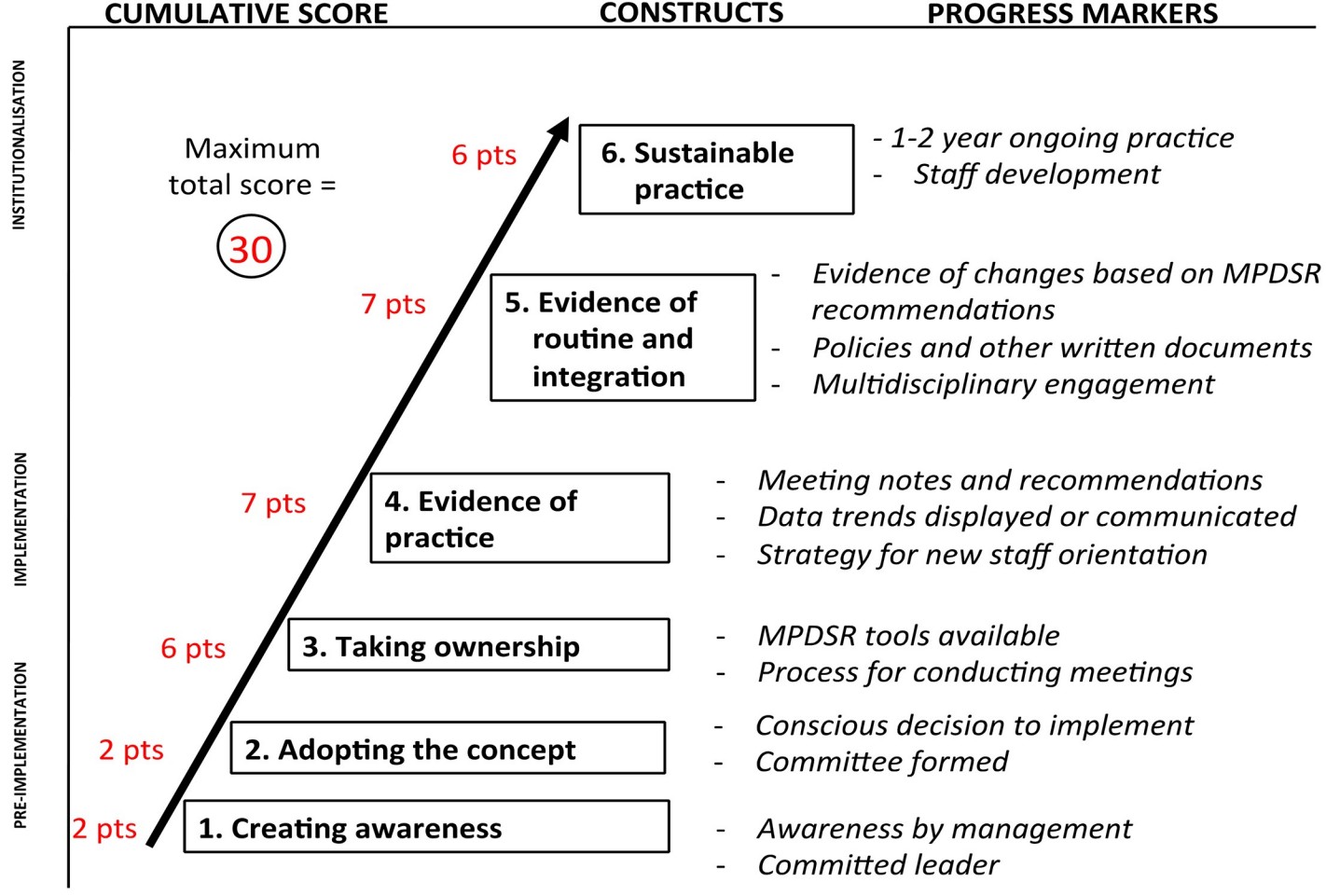

**Fig 1. Implementation progress scoring schematic.**

## Facility-based implementation of MPDSR

Across those facilities with evidence of practice, the stage of facility-based MPDSR implementation varied within and across countries (Fig 2). The mean implementation progress score across the 47 facilities was 18.98 (evidence of practice, Stage 4), with a range from 11.75–27.38. One-third of facilities (34%) had reached the evidence of practice stage (Stage 4); over half of facilities (55%) were assessed to be at the stage of routine and integrated practice (Stage 5); and 11% demonstrated implementation at the level of sustainable practice (Stage 6). Overall, hospitals scored higher on average (19.68) than health centres (16.01).

## Results by stage of facility-based MPDSR implementation

Results are reported for both specific progress markers and questionnaire items across stages that represent a linked implementation progression. Table 4 presents the results for all progress markers by individual country and cumulatively across the four countries. S4 Table provides the ranking of the progress markers by frequency overall. Progress markers for earlier stages (Stages 1–3) were mostly achieved by all facilities, which was consistent with facility selection criteria. Fewer facilities met the progress markers for higher stages of implementation

**FACILITY SCORES**  **STAGE OF CHANGE**  **NUMBER OF FACILITIES BY STAGE BY COUNTRY**

| | Nigeria (N=3) | Rwanda (N=13) | Tanzania (N=15) | Zimbabwe (N=16) | Total (N=47) |
|---|---|---|---|---|---|
| **MEAN SCORE BY COUNTRY** | | | | | |
| | 17.97 | 17.30 | 18.35 | 21.12 | **18.98** |

Cumulative score (out of 30)

| Stage of change | Nigeria (N=3) | Rwanda (N=13) | Tanzania (N=15) | Zimbabwe (N=16) | Total (N=47) |
|---|---|---|---|---|---|
| 6) Sustainable practice | - | 1 (8%) | - | 4 (25%) | **5 (11%)** |
| 5) Evidence of routine and integration | 2 (67%) | 6 (46%) | 10 (67%) | 8 (50%) | **26 (55%)** |
| 4) Evidence of practice | 1 (33%) | 6 (46%) | 5 (33%) | 4 (25%) | **16 (34%)** |
| 3) Taking ownership | | | | | |
| 2) Adopting the concept | | | | | |
| 1) Creating awareness | | | | | |

**Fig 2. Implementation progress score and distribution of facilities by country.**

(Stages 5 and 6), and wide variation was observed for some progress markers in the higher stages across countries (e.g., plans to ensure training). This section summarises results for each of the six stages of facility-based MPDSR implementation.

**Stage 1—Creating awareness.** The two progress markers for this stage were mostly achieved (by at least 68% of facilities). In most facilities (89%), leaders were fully involved in championing death audits, and nearly all facilities (98%) had a focal person responsible for conducting death audits. The individual assigned as the MPDSR coordinator varied by facility level. The facility in-charge was cited most commonly as the MPDSR coordinator in health centres and in small hospitals; the regional/district health officer for provincial, regional, and district hospitals; and the head of the obstetrics and gynaecology, paediatric, or neonatology department for tertiary and private hospitals. Introduction of MPDSR to facility staff varied by country and facility except in Rwanda, where respondents all reported a similar orientation process.

**Stage 2—Adopting the concept.** The two progress markers for this stage were mostly achieved. A 'formal decision to implement MPDSR' was recalled by facility staff in Nigeria, Rwanda, and Tanzania. However, some facility respondents in Zimbabwe could not recall the decision to begin implementing MPDSR. All facilities in Rwanda and Tanzania had established MPDSR steering committees, whereas only two of three facilities in Nigeria and 13 of 16 facilities in Zimbabwe had established committees.

**Stage 3—Taking ownership.** Among the seven progress markers in this stage, four were mostly achieved, one was moderately achieved (34–67% of facilities), and two were rarely achieved (< 33% of facilities), though findings varied among and within countries. Nearly all

**Table 4. Proportion of facilities meeting the progress markers for each stage of implementation (n = 47).**

| | Stage of implementation | Progress markers | Nigeria (n = 3) | Rwanda (n = 13) | Tanzania (n = 15) | Zimbabwe (n = 16) | Cumulative (n = 47) |
|---|---|---|---|---|---|---|---|
| **Pre-Implementation** | **1. Creating awareness (2 points)** | Awareness by management | 100%[c] | 100%[c] | 100%[c] | 94%[c] | 98%[c] |
| | | Committed leader | 100%[c] | 69%[c] | 100%[c] | 94%[c] | 89%[c] |
| | **2. Adopting the concept (2 points)** | Conscious decision to implement | 100%[c] | 100%[c] | 97%[c] | 84%[c] | 94%[c] |
| | | Committee formed | 67%[b] | 100%[c] | 100%[c] | 81%[c] | 91%[c] |
| **Implementation** | **3. Taking ownership (6 points)** | Tools available | 17%[a] | 100%[c] | 100%[c] | 69%[c] | 84%[c] |
| | | Tools include cause of death | 33%[a] | 100%[c] | 100%[c] | 63%[b] | 83%[c] |
| | | Tools include modifiable factors | 33%[a] | 100%[c] | 93%[c] | 72%[c] | 84%[c] |
| | | Tools include place to follow up on actions taken | 17%[a] | 100%[c] | 0%[a] | 59%[b] | 49%[b] |
| | | Understanding of process for conducting meetings | 100%[c] | 85%[c] | 93%[c] | 100%[c] | 94%[c] |
| | | Staff meeting conduct agreement available | 0%[a] | 8%[a] | 20%[a] | 0%[a] | 9%[a] |
| | | Budget or support to conduct death reviews | 100%[c] | 4%[a] | 10%[a] | 63%[b] | 32%[a] |
| | **4. Evidence of practice (7 points)** | Meeting minutes available | 50%[b] | 38%[b] | 87%[c] | 100%[c] | 74%[c] |
| | | Meeting minutes include action items | 17%[a] | 31%[a] | 100%[c] | 81%[c] | 68%[c] |
| | | Meeting minutes include follow-up from previous meetings | 17%[a] | 23%[a] | 20%[a] | 50%[b] | 30%[b] |
| | | Meeting notes respect confidentiality of staff and patients | 33%[a] | 31%[a] | 80%[c] | 97%[c] | 68%[c] |
| | | Face-to-face or written orientation to death reviews | 100%[c] | 92%[c] | 70%[c] | 53%[b] | 71%[c] |
| | | Data trends displayed or shared | 33%[a] | 50%[b] | 10%[a] | 41%[b] | 33%[a] |
| **Institutionalisation** | **5. Evidence of routine integration (7 points)** | Evidence of change based on recommendation | 61%[b] | 10%[a] | 44%[b] | 71%[b] | 44%[b] |
| | | Death review meetings are held at stated interval (e.g. weekly, monthly) | 67%[b] | 73%[b] | 47%[b] | 44%[b] | 53%[b] |
| | | Multidisciplinary engagement | 100%[c] | 85%[c] | 87%[c] | 91%[c] | 86%[c] |
| | | Evidence of reporting findings and progress to community | 17%[a] | 19%[a] | 37%[b] | 50%[b] | 34%[b] |
| | **6. Evidence of sustainable practice (6 points)** | Over 1–2 years of ongoing practice | 75%[c] | 85%[c] | 77%[c] | 95%[c] | 83%[c] |
| | | Plan in place to ensure all staff receive MPDSR training | 100%[c] | 0%[a] | 0%[a] | 53%[b] | 24%[a] |
| | | Evidence that staff have received MPDSR training in the past year | 67%[b] | 15%[a] | 63%[b] | 50%[b] | 45%[b] |

Note: The percentage provided signifies the number of facilities demonstrating the progress marker out of the total number with evidence of MPDSR practice.

[a] signifies "rarely achieved" and indicates less than 33% of facilities,

[b] signifies "moderately achieved" and indicates 34–67% of facilities, and

[c] signifies "mostly achieved" and indicates above 68% of facilities.

facilities (94%) could describe or show documentation of MPDSR processes. Standard MPDSR data collection forms were available in 84% of health facilities. Most facilities reported having a policy, guideline, or protocol available at the facility, which was shown to assessors, and for the most part, it was the national guideline. Nigeria was the exception, as facilities reported no written MPDSR policy, guidelines, or tools available in the facility. MPDSR tools included cause of death and modifiable factors in facilities in Rwanda, Tanzania, and Zimbabwe. Most facility tools across the four countries lacked a designated place to document

follow-up on actions taken (i.e., response), except for in Rwanda, where the standard MPDSR form includes a place to document follow-up of actions. There was strong awareness of national MPDSR guidelines among facility interviewees in Rwanda and Zimbabwe. Few of the facilities in Nigeria were aware of the national guidelines. In Tanzania, all facilities were aware of the national guideline, but five hospitals demonstrated gaps in adhering with the national guideline, notably around information flow to other levels and community follow-up. Respondents at both the facility and subnational levels described how they valued the process of reviewing cases:

'*Providing information about preventable factors that contribute to maternal death and using information to guide actions is key for preventing similar death in the future*'.

–*Facility interview, Rwanda*

'*We may think it's too much to review every death, but each one death is crucial to someone. It might be a statistic to me, but every death matters*'.

–*Stakeholder interview, Zimbabwe*

Few facilities had agreements or procedures in place regarding the conduct of MPDSR meetings (9%). Nearly one-quarter of facilities (23%) reported a connection between professional disciplinary actions and MPDSR activities, including one facility in Rwanda, three in Tanzania, two in Nigeria, and six in Zimbabwe. In Nigeria, only one of three facilities reported a nonpunitive, no-blame environment. Respondents described different approaches to assigning blame within MPDSR activities:

'*Review meetings are where people learn to "stick to the rules". . . . Some staff are reprimanded verbally and [receive] other punishments*'.

–*Facility interview, Nigeria*

'*The health worker involved is requested to provide a statement of how the incident happened and may be given a verbal warning or a written one. . . and in one incident, the responsible person did not work for 1 month*'.

–*Facility interview, Tanzania*

One-third of all facilities reported financial or in-kind support from the hospital budget or partner allocations to establish or support MPDSR activities. Hospital or district budget support to establish MPDSR processes varied starkly across facilities, ranging from 15% of facilities in Rwanda, to 33% of facilities in Nigeria and Tanzania, to 69% of facilities in Zimbabwe.

**Stage 4—Evidence of practice.**    Four of the six progress markers were mostly achieved in this stage. Minutes of MPDSR meetings were observed in 74% of facilities; meeting minutes included action items and respected the confidentiality of staff and patients in two-thirds (68%) of facilities. One-third of facilities (30%) presented meeting minutes with documented follow-up of prioritised actions from previous meetings. Qualitative interviews emphasised the importance of meeting minutes and written recommendations:

'*We need to document the meetings better with minutes and give the designated actions to the responsible persons in writing*'.

–*Facility interview, Tanzania*

'*One of the most challenging parts of the review process is the formulation of appropriate recommendations, but this step is critical to successful MPDSR*'.

*–Facility interview, Rwanda*

Overall, 71% of facilities provided some sort of orientation on MPDSR to facility staff members, ranging from 53% of facilities in Zimbabwe to 100% in Nigeria. The assessment did not explore who attended orientations, how an orientation was conducted, or why one was not conducted.

Only one-third of facilities demonstrated the display or sharing of data trends (e.g., run charts with key statistics posted on a wall). The most commonly mentioned sources of data on death were the labour and delivery registers, followed by the postnatal register. At facilities responsible for capturing information on maternal and perinatal deaths in the community (four of six health centres in Tanzania, nine of 16 facilities in Zimbabwe, and three of 13 facilities in Rwanda), assessors observed gaps in the information provided in the case files. Data sources for compiling case reports in advance of death audit meetings included patient clinical records, registers, transfer/referral forms, and ambulance records. Guidance on methods to classify deaths varied from an optional checklist approach, to open-ended questions on apparent causes of death, to ICD-10 classification (The 10th revision of the *International Statistical Classification of Diseases and Related Health Problems* [ICD-10]). Less than one-half of the facility respondents (47%) reported that the medical records and registers captured the information necessary to determine cause of death and identify contributing factors (ranging from 27% of facilities in Tanzania to 75% of facilities in Zimbabwe). Cause of death classification systems varied among and within countries. Two-thirds of facility respondents reported using some form of standard coding system aligned with the national guideline on the mortality audit forms (66%). For modifiable factors, almost all facilities reported classifying deaths as avoidable, possibly avoidable, or not avoidable, and/or used the three delays model or a root cause analysis [35]. Facility respondents expressed varying perceptions of the accuracy of data:

'*One cannot vouch for the accuracy of data being collected because staff are not motivated. They do not know what it will be used for*'.

*–Facility interview, Nigeria*

'*I strongly believe the forms provide adequate information, but the big challenge here resides in providers who do not fill in the necessary information. In general, information is not filled in the forms*'.

*–Stakeholder interview, Zimbabwe*

'*We always need to reconcile the cause of death data from the MPDSR form and register to avoid discrepancies of deaths in facilities*'.

*–Facility interview, Tanzania*

**Stage 5—Evidence of routine and integrated practice.** Only one of the four progress markers in this stage (multidisciplinary engagement) was mostly achieved in at least two-thirds of facilities, while the other three progress markers were only moderately achieved. Most facilities reported that they assigned specific follow-up actions to individuals with timelines (79%). Less than one-half of the facilities (44%) could actually demonstrate or show any evidence of

change(s) made based on recommendations from death reviews (Fig 3). Examples of changes described by facility respondents included improved clinical practices, referrals, documentation, and procurement of essential commodities (e.g., blood). The quote below by a facility respondent provides an example of a successful local response:

> '*Now that the perinatal death is audited, they have started resuscitation of babies who are not crying or breathing. Also, proper use of partographs is now in place*'.
>
> –*Facility interview, Tanzania*

Though national guidelines included schematics on the reporting structure, including how responses should be tracked, less than one-third (28%) of facilities reported a formal written documentation system for tracking follow-up of recommended actions. Only one facility each in Zimbabwe and Tanzania and three in Rwanda demonstrated a formal process for follow-up of recommendations, apart from reviewing minutes at the next mortality audit meeting. None of the facilities in Nigeria had a systematic process for following up on recommendations.

One-half of facilities held meetings on a predetermined schedule (53%), ranging from 47% in Zimbabwe to 73% in Rwanda. Other facilities held meetings only after a death occurred or on an ad hoc basis. The reporting of regular MPDSR meetings by facility respondents was generally greater than observable evidence of regular meetings (e.g., through review of meeting minutes).

Most facilities demonstrated evidence of multidisciplinary participation in death audit meetings (86%) with representation of a range of health workers from different units,

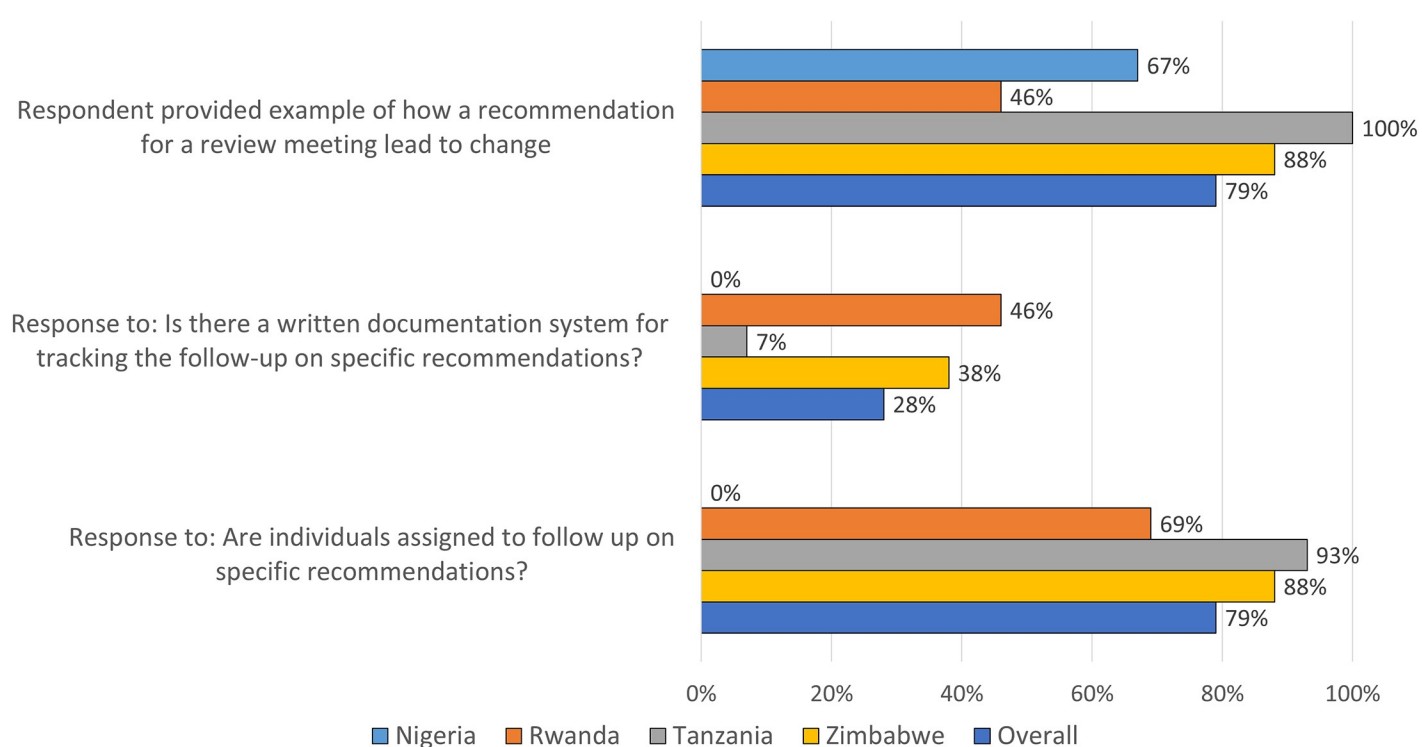

**Fig 3. Proportion of facilities reporting follow-up of recommended actions from death reviews (N = 47 facilities).**

especially in larger facilities. Respondents explained the value of the multidisciplinary nature of the meetings and some of the challenges posed around attendance given staff shortages.

> '*Everyone attends our maternal and perinatal meetings, all the way to the driver, because when we have a case to transfer, he knows why we need to move now*'.
>
> –*Facility interview, Zimbabwe*

> '*It's helping [the MPDSR process]. One person wouldn't have noted these gaps alone. But together, we are improving the quality of services*'.
>
> –*Facility interview, Zimbabwe*

> '*There are not enough staff to attend meeting as well as tend to patients*'.
>
> –*Facility interview, Tanzania*

> '*We have difficulty finding an opportunity to gather everyone due to busy schedules*'.
>
> –*Facility interview, Nigeria*

Three-quarters of health facilities reported regularly linking MPDSR to other quality improvement activities at their facilities (74%). However, none of the national guidelines included clear guidance on linking MPDSR to quality improvement activities, and the team did not systematically assess the linkages.

One-third of the facilities reported sharing death audit findings, recommendations, and progress with the community (34%), including four facilities in Rwanda, seven in Zimbabwe, and two in Tanzania (none in Nigeria). The reported channels of communication varied among and within countries. Audit recommendations were typically shared with community health workers to disseminate to the community in Rwanda, whereas in Zimbabwe, some facilities reported that a facility staff member was designated as a community liaison and was responsible for sharing recommendations with the community. One facility respondent in Tanzania reflected the desire to provide feedback but did not have a mechanism to do so, a sentiment echoed by other facilities:

> '*We wish that there was a specific mechanism to ensure that MPDSR feedback is shared with the community*'.
>
> –*Facility interview, Tanzania*

**Stage 6—Evidence of sustained practice.** The three progress markers in this stage ranged from rarely achieved to mostly achieved. Most facilities assessed (83%) achieved the progress marker for demonstrating occurrence of death audit meetings for at least 1 year (irrespective of regularity). Evidence of staff development to sustain MPDSR practice was partially achieved, with only 45% of facilities reporting that staff had received MPDSR training in the past year. A plan in place to ensure all staff receive MPDSR training was rarely achieved by the assessed facilities (24%), with no future plans observed at the facilities in Rwanda and Tanzania. The qualitative responses supported these findings:

> '*By policy, the ward in-charge is supposed to be trained in MPDSR, but she has not had any training, even though she is preparing the case summary*'.
>
> –*Facility interview, Tanzania*

## Enablers and barriers to MPDSR

Table 5 summarises the top three barriers and enablers of MPDSR implementation as observed by the assessors and as reported by facility informants. The top three enablers observed by the assessors included leadership, regular meeting conducted with participation from a multidisciplinary team, and availability and use of the MPDSR-related guidelines and tools. The top three barriers observed by the assessors included lack of health worker capacity to capture and use data analytically to inform the review process, limited plans for training health workers on the MPDSR process, and limited accountability for the follow-up actions identified during the review process. S3 Table provides detailed results of the identified MPDSR implementation enablers and barriers analyses by country.

The most commonly described enabling factors by informants across countries included teamwork, communication between staff, staff commitment, and multidisciplinary participation during meetings. Other reported enablers across the countries included national and subnational support through MPDSR training support and evidence of MPDSR process leading to change or having improved health services. Additional cited enablers included availability of MPDSR guidelines and tools, facility leadership for MPDSR, observed positive effect of MPDSR process on reducing deaths, and staff motivation to support MPDSR due to concern about high number of deaths. The most commonly cited barriers to implementing MPDSR processes described by facility staff included limited staff time, heavy workloads preventing participation in meetings, general staff shortages, and high staff turnover. Other reported barriers included lack of motivation due to absence of incentives for participation in meetings (e.g., travel support) or perceived lack of effect of death audit meetings (e.g., audit recommendations not implemented, health services unchanged.) The most commonly cited changes to improve the utility of MPDSR included actions to motivate staff, such as providing incentives for participation in MPDSR processes, increasing facility staff numbers, increasing MPDSR capacity and skills through additional training and mentorship, more funding and specific resources to facilitate meeting and data collection processes, stronger facility leadership of MPDSR, more regular death review meetings, multidisciplinary participation, and reducing the blame environment.

**Table 5. Top enablers and barriers to MPDSR implementation.**

| Top three enablers | Top three barriers |
|---|---|
| *Based on observations* | |
| Leadership by individual(s) in promoting death reviews including management, professionals, driving forces | Lack of health worker capacity to capture and use data analytically to inform the review process |
| Regular meeting conducted with participation from a multidisciplinary team | Limited plans for training health workers on the MPDSR process |
| Availability and use of the MPDSR-related guidelines and tools | Limited accountability for the follow-up actions identified during the review process |
| *Based on response from the facility informants* | |
| Interdisciplinary teamwork with good communication amongst staff and staff participation in meetings | Health worker capacity issues, such as limited staff time and work overload, preventing meeting attendance |
| Support from national and/or subnational levels, including through training, capacity-building, and administrative support | Human resource shortage issues, such as high staff turnover and general staff shortage |
| Evidence of MPDSR process leading to change or having improved health services | Demotivation due to recommendations at various levels not being implemented |

## Discussion

This assessment of MPDSR implementation aimed to characterise the stages of MPDSR implementation progress across several countries using a standardised scoring methodology. The assessment results reinforce previous findings [17,21–23] and highlight important implementation gaps and priority areas to strengthen MPDSR systems in low-capacity settings.

### Implementation factors

A supportive policy and political environment for MPDSR facilitates implementation but does not guarantee translation into practice [22,23,32,33]. Components in national guidelines that are more straightforward to implement, such as establishment of a steering committee or assigning an MDSR or perinatal death surveillance and response coordinator, generally had greater uptake in facilities. Components of the national guidelines with fewer details (e.g., cause of death classification, or follow-up on action plans or community linkage) demonstrated more variable practice across facilities. Ensuring onsite availability of practical guidance and tools is a critical component at the pre-implementation phase [21]. The history of MPDSR introduction and implementation also matters for sustaining and institutionalising MPDSR practice [24,36], as demonstrated by Zimbabwe, which had the highest overall score (27.38) and has a long history of practising MPDSR in central-level hospitals. While the national guidelines could be strengthened in some areas, such as not having clear instructions on how to follow up on the recommendations, they were mostly aligned with the WHO global guidelines and all had useful tools for implementation, which would enable a supportive policy and political environment to initiate and support implementation [33]. The primary challenge of implementation appears to be at the organizational and individual levels, which are the coalface of implementation [33].

This study confirmed previously reported common facilitators of MPDSR, including the importance of strong leadership and effective teamwork [21–24,37–41]. Engagement of subnational managers promotes accountability and supports MPDSR practice at facility level through cross-facility/-district learning, capacity-building, and mentorship [24,33,40]. Multifaceted efforts to improve quality of care, including MPDSR, emphasise leadership and teamwork, understanding of the root causes of local quality of care gaps, and the systematic implementation of changes to close gaps [23,32,33,42]. There are many opportunities to strengthen alignment of broader quality improvement and MPDSR processes. For example, MPDSR generates essential information about the local causes of maternal and perinatal deaths and the key contributors to these deaths, which is important for designing robust quality improvement efforts that are responsive to local needs. Quality improvement efforts typically include a systematic change management and monitoring strategy. They can help bolster the systematic follow-up and measurement of the effect of death audit recommendations, an area of weakness identified in this assessment.

Linked to teamwork, the organisational culture around the death audit process can either facilitate or inhibit implementation of MPDSR. Previous studies have found that a lack of trust between health professionals and service administrators, a culture of blame and fear of potential legal ramifications, and the lack of ownership of a process prevent successful implementation [22,32,43]. Failure to comply with principles of confidentiality and anonymity can inhibit implementation practice [22,23,32,41,43–46]. A culture of safety in which staff feel protected from disciplinary action and in which death audit data are de-identified and/or kept confidential is a WHO-recommended practice [19,20]. If staff fear repercussions, they are unlikely to support MPDSR or engage fully and productively in an audit process. Elements of individual-level fault-finding and/or disciplinary processes were reported in one-quarter of the facilities

in this study, though comments made by respondents during the interview process suggested blame and disciplinary action occurred more than was reported. A study in Nigeria found that the interactional processes among those involved in audit meetings affect the meaningfulness of the death review and may inhibit their impact [34]. Deeper investigation is needed to better characterise and understand the impact that a 'blame culture' has on the effectiveness of the MPDSR process. Strategies, such as official audit charters or codes of conduct that are mentioned in the national guidelines, may minimize acrimony and prevent (or reduce) blame and recriminations [47,48]. Few facilities in this assessment had formal agreements or procedures in place regarding the conduct of MPDSR meetings despite facility staff undergoing some type of training or having access to guidelines, which made this recommendation.

Poor staff motivation, limited time and capacity, poorly functioning health systems, and general human resource challenges have also been shown to undermine MPDSR efforts [25,36,37,44,49,50]. Success of MPDSR relies on an individual's and team's willingness to 'self-correct'; commit to honest, open discussions with peers about a traumatic event; and implement recommended actions [33]. When problems identified during review meetings are not followed up on and addressed, staff lose motivation to participate in MPDSR activities [22,34,51,52]. At the facility level, this assessment demonstrated a lack of consistent follow-up of recommended actions and infrequent sharing of success stories arising from the audit process. Further investigation is needed to determine how this affects the motivation of facility staff.

Prior studies demonstrate that the confidence and capability of health workers to complete the review process and analyse death audit data strongly influence implementation of effective MPDSR processes [21,23,24,32,36,41,49,52,53]. Low confidence of managers and health workers to assess causes of deaths and modifiable factors documented in this assessment confirm the findings of prior studies and illustrate the importance of strengthening health worker confidence, skills, and information systems to support MPDSR. Several studies have shown that stronger health information systems, including improved data capture, use, and reliability, can facilitate MPDSR processes [23,32,36–38,40,45,47,52]. The common lack of mortality and patient care data in routine health information systems in low-resource settings (e.g., patient records/case notes, facility registers) hinders robust MPDSR implementation, including accurate assignment of cause of death and identification of critical gaps in quality of care [42]. In this assessment, subnational managers expressed concern about the quality of data in facility MPDSR reports, and less than one-half of facility respondents reported that the health information available in their facility was sufficient to classify cause of death and analyse contributing factors. None of the national guidelines in the four assessment countries explicitly aligned with the WHO ICD-10 maternal mortality guidelines [54], published before the most recently updated guidelines in each country, nor the WHO ICD-10 perinatal mortality guidelines, published at the time of the assessment [55]. There is a need to strengthen health information systems and assignment of cause of death guidance in both policy and practice.

Reliance on external funds and/or goodwill of professional organisations to support administration, training, and implementation of MPDSR processes have previously been identified as a barrier to sustainable practice [23,47,56,57]. It is unclear whether designated funding (e.g., a budget line item) is important for effective MPDSR implementation. This assessment did not demonstrate a close relationship between reported budgetary or in-kind support and facility conduct of death audits. Presence of donor support in some areas may have boosted findings of sustainable practice but this would need to be investigated further.

Community engagement may strengthen collective ownership, responsibility (e.g., for referral), and quality of maternal and perinatal care, and may contribute to more robust implementation of MPDSR processes [21,22,32,44,57,58]. The small proportion of facilities

reporting sharing death audit recommendations with the community in all four countries deserves greater exploration. Learning from studies of facilities undertaking intentional efforts to engage communities should be further explored to determine how such community engagement might influence the accountability mechanism of death audits and how this may influence community behaviours [22,59].

## Measuring implementation

This assessment was the first to our knownledge to apply a standardised implementation progress scoring model to assess MPDSR implementation. The related tool developed for the assessment sought to classify progress markers of MPDSR processes derived from the literature. Its sensitivity in being able to correctly identify a facility's ability to demonstrate specific implementation markers could not be formally assessed in comparison to alternative tools for MPDSR since it was the first of its kind. The progress markers measure the current status of implementation, especially in terms of tangible and immediate indicators of organizational commitment to implement MPDSR processes including committees formed, training, focal point identified, and availability of tools. It is important to note, however, that the tool was not designed to assess the quality of specific MPDSR processes (e.g. correct assignment of causes of death; robust identification of modifiable contributors to deaths audited; development and follow up of actionable responses to address identified contributors, ability to correct mismanagement etc. . .). Future applications of this standardised implementation progress scoring model methodology for MPDSR should review the stage-specific progress markers, data collection tools, and process of assigning a standardised implementation score based on learnings from this assessment. Additional progress markers of implementation coverage, such as proportion of deaths reviewed based on national recommendations, should also be considered. Clear operational definitions for each marker will strengthen inter-rater reliability and systematic measurement across sites.

## Limitations

The assessment was conducted in a relatively small number of nonrandomly selected facilities in only four countries; therefore, it is not possible to generalise the assessment findings at the country subnational or national level or for the continent of Africa. Given the purposeful, non-representative sample of facilities, the team was not able to analyse potential patterns or differences in MPDSR implementation by facility type (e.g., rural versus urban, primary versus secondary). The nature of the study is a source of possible biases [60]. First, the choice of facilities was made on the basis of a specific program favouring MPDSR. Second, interviews were led by people who may have had an interest in presenting the program in a favourable light. Third, the assessors had a background in clinical care for maternal and newborn health and/or worked for non-governmental organizations, professional associations, or Ministry of Health bringing their own professional background, experiences and prior assumptions. Power dynamics between assessors and those interviewed may have impacted on participants' willingness to talk openly about experiences. Despite efforts to standardise data collection across countries, the variation in individual assessors and the modest adaptation of data collection tools in each country may have also contributed to some variation in the scoring approach in individual facilities and countries. Data were collected from health workers present at the facility on the specific day of the facility visit; thus, the views and MPDSR activities reported by facility respondents may not capture all facility-specific MPDSR activities or reflect the views of all health care staff, including junior staff, who may be subject to more blame or scrutiny during mortality audit meetings and who may have been absent on the day of the assessment

or more hesitant to share their views during group interviews. The non-availablity of subnational stakeholders in Rwanda at the time of the assessment is another limitation to note.

For the most part, this assessment did not differentiate between maternal and perinatal death audit processes. Further research is needed to distinguish differences in death audits and responses for maternal and perinatal deaths. The study included both health centres and hospitals but was not designed to investigate differences in implementation between the two different levels. Further research is needed to explore characteristics of implementing MPDSR in a health center versus a hospital setting.

The assessment set out to measure implementation status and did not evaluate the quality of MPDSR processes (e.g., surveillance completeness, accuracy of cause of death assignment, analysis of modifiable factors, development and follow-up of actions).

## Conclusion

This assessment is the first attempt, to the authors' knowledge, to assess facility-level MPDSR implementation progress using a standardised scoring methodology in multiple countries. Structures and processes for implementing MPDSR existed in all four countries, with over two-thirds of the assessed facilities reaching at least stage 5 –evidence of routine and integrated practice. Many implementation gaps were identified that can inform priorities for strengthening MPDSR implementation. These gaps include ensuring availability of onsite MPDSR guidelines and forms, developing more explicit guidance on cause of death assignment and follow-up of audit recommendations across system levels as part of national guidelines, instituting regular mechanisms to build manager and health worker confidence and skills to implement MPDSR (e.g., training, supervision), strengthening health information systems to permit accurate classification of cause of death and support robust death reviews, strengthening alignment of MPDSR and broader quality improvement efforts, and increasing linkages across system-level MPDSR activities, from community, to facilities, to regional and district health managers. Further implementation research is needed to assess the quality of MPDSR implementation processes and to identify and test mechanisms to overcome common MPDSR implementation gaps in low-capacity settings.

## Supporting information

**S1 Table. Data sources and collection methods.**
(DOCX)

**S2 Table. Mapping content of national MPDSR policy by country.**
(DOCX)

**S3 Table. Summary of MPDSR implementation enablers and barriers most commonly cited by facility staff in four countries.**
(DOCX)

**S4 Table. Ranking of progress markers by frequency across 47 facilities.**
(DOCX)

**S1 File. Brief historical summary of MPDSR processes by country.**
(DOCX)

**S1 Data. Database for MCSP multicountry assessment of MPDSR implementation.**
(DOCX)

## Acknowledgments

We would like to thank the ministry of health officials, USAID Missions, professional association members, and health workers from Nigeria, Rwanda, Tanzania, and Zimbabwe for their input and support. A very special thanks to many in-country MCSP staff and technical experts who supported the research implementation process. In addition, we would like to thank, Lara Vaz (Save the Children US), Stella Abwao (MCSP), John Varallo (MCSP), Lisa Noguchi (MCSP), Ayne Worku (MCSP), and Brianne Kallam (MCSP) for their technical input and review of drafts of the individual country and multicountry reports. We appreciate the support from Edward Kenyi (MCSP) on data extraction. We thank Sylvia Alford with USAID for her strong support during all stages of planning and implementing this assessment and analyzing and disseminating assessment results.

## Author Contributions

**Conceptualization:** Kathleen Hill, Kate Kerber.

**Data curation:** Mary V. Kinney, Gbaike Ajayi, Alyssa Om'Iniabohs, Kate Kerber, Kusum Thapa.

**Formal analysis:** Mary V. Kinney, Gbaike Ajayi, Joseph de Graft-Johnson, Kusum Thapa.

**Funding acquisition:** Kathleen Hill, Neena Khadka.

**Investigation:** Mary V. Kinney, Gbaike Ajayi, Joseph de Graft-Johnson, Kathleen Hill, Neena Khadka, Alyssa Om'Iniabohs, Fadzai Mukora-Mutseyekwa, Edwin Tayebwa, Oladapo Shittu, Chrisostom Lipingu, Kate Kerber, Juma Daimon Nyakina, Perpetus Chudi Ibekwe, Felix Sayinzoga, Bernard Madzima, Kusum Thapa.

**Methodology:** Gbaike Ajayi, Joseph de Graft-Johnson, Kathleen Hill, Neena Khadka, Kate Kerber, Kusum Thapa.

**Project administration:** Gbaike Ajayi, Alyssa Om'Iniabohs.

**Supervision:** Mary V. Kinney, Joseph de Graft-Johnson, Kathleen Hill, Neena Khadka, Fadzai Mukora-Mutseyekwa, Edwin Tayebwa, Oladapo Shittu, Chrisostom Lipingu, Kate Kerber, Felix Sayinzoga, Asha S. George, Kusum Thapa.

**Validation:** Mary V. Kinney, Kathleen Hill, Neena Khadka, Kate Kerber, Asha S. George, Kusum Thapa.

**Visualization:** Mary V. Kinney, Gbaike Ajayi, Alyssa Om'Iniabohs, Kate Kerber, Asha S. George, Kusum Thapa.

**Writing – original draft:** Mary V. Kinney, Gbaike Ajayi, Kusum Thapa.

**Writing – review & editing:** Mary V. Kinney, Gbaike Ajayi, Joseph de Graft-Johnson, Kathleen Hill, Neena Khadka, Alyssa Om'Iniabohs, Fadzai Mukora-Mutseyekwa, Edwin Tayebwa, Oladapo Shittu, Chrisostom Lipingu, Kate Kerber, Juma Daimon Nyakina, Perpetus Chudi Ibekwe, Felix Sayinzoga, Bernard Madzima, Asha S. George, Kusum Thapa.

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
