## [Decision Letter · Decision Letter 0]

29 Apr 2020

PONE-D-20-00503

“It might be a statistic to me, but every death matters.”: An assessment of facility-level maternal and perinatal death surveillance and response systems in four sub-Saharan African countries

PLOS ONE

Dear Ms Kinney,

Thank you for submitting your manuscript to PLOS ONE. After careful consideration, we feel that it has merit but does not fully meet PLOS ONE’s publication criteria as it currently stands. Therefore, we invite you to submit a revised version of the manuscript that addresses the points raised during the review process.

We would appreciate receiving your revised manuscript by Jun 12 2020 11:59PM. To enhance the reproducibility of your results, we recommend that if applicable you deposit your laboratory protocols in protocols.io, where a protocol can be assigned its own identifier (DOI) such that it can be cited independently in the future. For instructions see: http://journals.plos.org/plosone/s/submission-guidelines#loc-laboratory-protocols

We look forward to receiving your revised manuscript.

Kind regards,

Natasha McDonald

Associate Editor

PLOS ONE

Journal Requirements:

Please explain why was written consent was not obtained, how you recorded/documented participant consent, and if the ethics committees/IRBs approved this consent procedure.

Reviewers' comments:

Reviewer's Responses to Questions

**Comments to the Author**

1. Is the manuscript technically sound, and do the data support the conclusions?

Reviewer #1: Yes

Reviewer #2: Partly

2. Has the statistical analysis been performed appropriately and rigorously? 

Reviewer #1: N/A

Reviewer #2: Yes

3. Have the authors made all data underlying the findings in their manuscript fully available?

Reviewer #1: Yes

Reviewer #2: Yes

4. Is the manuscript presented in an intelligible fashion and written in standard English?

Reviewer #1: Yes

Reviewer #2: Yes

5. Review Comments to the Author

Reviewer #1: General comment

Assessing the implementation of the maternal and perinatal death surveillance and response system (MPDSR) is a relevant research topic. Indeed, there are very few serious evaluations of the implementation of such MPDSR systems. The strength of this paper, apart that it is beautifully written, is the trial to set up a standardized assessment tool based on scores for each of the six stages of implementation defined by the authors. The weaknesses are linked to the biased sample of facilities investigated, the debatable choice of markers in the scores, the small scope of the literature reviewed for the topic, and the relative lack of investigation of the factors enabling the implementation or the barriers. Nevertheless, I found the paper sufficiently interesting to be published. I have a few suggestions below.

Detailed comments

The introduction is excellent: well organized and well written.

Method section

The sample is purposive, depending on the presence of US Agency for International Development (USAID)’s Maternal and Child Survival Program (MCSP) staff. This is of course a source of two possible biases: 1) a choice of facilities made on the basis of a specific program favouring MPDSR and 2) interviews led by people who have an interest in making the program a success (there was no allusion to how the researchers dealt with reflexivity, preconceptions and metapositions (Malterud K. 2001. Qualitative research: standards, challenges, and guidelines. Lancet 358: 483–88)).

The authors described the number of facilities selected in each area but did not provide the total number of facilities in these areas. That would give some perspective.

There is a mixture of health centres and hospitals in the sample. Why health centres? Do they all implement death reviews? Are health centres equipped to deal with severe morbidity? Are these patients not transferred to hospitals?

In Rwanda, no stakeholder was interviewed. Why? Is it really only a question of availability?

Table 3 (and Figure 3) is quite interesting since the authors described the rationale or their hypotheses on which the markers are based, and allows some discussion. Did they try a sensitivity analysis to estimate the effect of modifying a score?

Line 140: how many assessors for the same facility? Did the authors find any discrepancies between the data collectors?

Line 146: the fact that facilities with a score of less than 10 points were excluded clearly means that only those facilities with some success in implementation are investigated. What is the reason explaining why these facilities did not reach the minimum score of 10 points? What is then the meaning of an average score (very precise!) calculated only for the best implementers (line 173)?

I wonder why in the progress markers there was nothing about the coverage of deaths reviewed, i.e. the number of maternal deaths reviewed on the total of maternal deaths during the period and the same for perinatal deaths. This may be a well standardized indicator useful for comparing facilities if the policy is to review each case, of course. A second indicator would be the capacity of a facility to stop/prevent the problems/mismanagement that contributed to the deaths.

Results

Table 4 is not easy to understand. Actually, each stage seems independent from the others while in reality it is difficult to understand that a facility without meeting minutes available, action items and follow-up have regular meetings and engagement.

To come back on the suggestion of building an indicator that identifies the progress in correcting mismanagement, the verbatim line 227 (‘Providing information about preventable factors that contribute to maternal death and using information to guide actions is key for preventing similar death in the future.’ –Facility interview, Rwanda’) shows that this is also a demand from the health workers and something measurable.

It is surprising to read that line 232: Few facilities had agreements or procedures in place regarding the conduct of MPDSR meetings (9%). All teams were supposed to have been trained. This should be discussed.

Line 298: “Though national guidelines included schematics on the reporting structure, including how responses should be tracked, less than one-third (28%) of facilities reported a formal written documentation system for tracking follow-up of recommended actions. Only one facility each in Zimbabwe and Tanzania and three in Rwanda demonstrated a formal process for follow-up of recommendations, apart from reviewing minutes at the next mortality audit meeting. None of the facilities in Nigeria had a systematic process for following up on recommendations.” This seems to be an interesting synthesis indicator of the effectiveness of maternal and perinatal death reviews.

I appreciated also the ST4 with a trial to build a full synthesis indicator that considers each successive step. The progress marker ‘There is evidence of change based on recommendations that arise from death review findings’ reached by 45% of facilities is an achievement, even in a sample of facilities supported by the USAID program.

The section of enablers and barriers is really small and does not add to what we already know.

Discussion

The discussion section does not discuss the limitations of the method itself. To what extent the scoring sufficiently reflects the effectiveness of the implementation of these death reviews?

The discussion section is supported by few references (8x #22, 7x #23, 4x #32 and 9 other references), see below ‘supporting literature’.

Supporting literature

Most of the cited literature are documents or articles from WHO or WHO civil servants, then from US authors. Table 3, explaining the rationale for the items in the score, is mainly based on 3 references (#21 13x; #22 5x; and #23 5x) while the discussion is based mainly on 2 references (#22 8x and #23 7x).

However, important literature on clinical audits and maternal death reviews comes from different groups of researchers. Among them: Johnston G, Crombie I K, Alder E M, Davies H T O and Millard A. Reviewing audit: barriers and facilitating factors for effective clinical audit Qual. Health Care 2000;9;23-36. doi:10.1136/qhc.9.1.23; Ivers N, Jamtvedt G, Flottorp S, Young JM, Odgaard-Jensen J, French SD, O’Brien MA, Johansen M, Grimshaw J, Oxman AD. Audit and feedback: effects on professional practice and healthcare outcomes. Cochrane Database of Systematic Reviews 2012, Issue 6. Art. No.: CD000259. DOI: 10.1002/14651858.CD000259.pub3; Filippi V, Brugha R, Browne E, Gohou V, Bacci A, De Brouwere V, Sahel S, Goufodji S, Alihonou E, Ronsmans C. 2004. How to do (or not to do) . . . Obstetric audit in resource poor settings: lessons from a multi-country project auditing ‘near miss’ obstetrical emergencies. Health Policy and Planning, 19(1), 57-66; Müffler N, Trabelssi M, De Brouwere V. 2007. Scaling up clinical audits of obstetric cases in Morocco. Tropical Medicine & International Health 12(10), 1248-1257; and the numerous papers from Kongnyuy et al. exploring maternal death reviews in Malawi).

Please, note that the references are not standardized: sometimes, the reference is with the acronym of the journal (bjog), sometimes in full (Health policy & planning) and careful attention should be paid to have a correct list.

Reviewer #2: The rate of maternal and perinatal death in developing countries are still high, so it is urgent issue to prevent future deaths. Therefore, this study is very significant. A huge amount of data has been analyzed in detail and I think it is valuable information.

Please tell me about the following.

1) It was written that you used a standardized scoring methodology. Which is the name of the score scale and which paper shows that the scale is standardized?

Is there a description of the validity of this measure in this paper?

2) What is the mean implementation progress score in developed countries, especially those with low maternal and perinatal mortality?

3) It feels redundant overall, can you summarize it a little more concisely? You can leave the necessary information as it is.

6. PLOS authors have the option to publish the peer review history of their article (what does this mean?). If published, this will include your full peer review and any attached files.

Reviewer #1: Yes: Vincent De Brouwere

Reviewer #2: No

---

## [Author Response · Author response to Decision Letter 0]

7 Jun 2020

Comments to the Author - RESPONSE FROM AUTHORS IN CAPS

REVIEWER #1 

Assessing the implementation of the maternal and perinatal death surveillance and response system (MPDSR) is a relevant research topic. Indeed, there are very few serious evaluations of the implementation of such MPDSR systems. The strength of this paper, apart that it is beautifully written, is the trial to set up a standardized assessment tool based on scores for each of the six stages of implementation defined by the authors. The weaknesses are linked to the biased sample of facilities investigated, the debatable choice of markers in the scores, the small scope of the literature reviewed for the topic, and the relative lack of investigation of the factors enabling the implementation or the barriers. Nevertheless, I found the paper sufficiently interesting to be published. I have a few suggestions below.

THANK YOU FOR YOUR HELPFUL REVIEW. WE HAVE ADDRESSED THE WEAKNESSES MENTIONED HERE IN THE RESPONSES BELOW AND IN THE MANUSCRIPT. 

Detailed comments

The introduction is excellent: well organized and well written. THANK YOU.

Method section

The sample is purposive, depending on the presence of US Agency for International Development (USAID)’s Maternal and Child Survival Program (MCSP) staff. This is of course a source of two possible biases: 1) a choice of facilities made on the basis of a specific program favouring MPDSR and 2) interviews led by people who have an interest in making the program a success (there was no allusion to how the researchers dealt with reflexivity, preconceptions and metapositions (Malterud K. 2001. Qualitative research: standards, challenges, and guidelines. Lancet 358: 483–88)).

THANK YOU FOR NOTING THE ISSUE OF REFLEXIVITY AND THE IMPORTANCE OF ACKNOWLEDGING ANY POTENTIAL BIASES. WE HAVE INCLUDED A STATEMENT IN THE LIMITATION SECTION NOTING THESE POTENTIAL BIASES.

The authors described the number of facilities selected in each area but did not provide the total number of facilities in these areas. That would give some perspective.

THANK YOU FOR THIS RECOMMENDATION. GIVEN THAT THE FACILITIES IN THE ASSESSMENT WERE SELECTED FROM THOSE PROVIDING CHILDBIRTH SERVICES AND WITH CURRENT OR RECENT EXPERIENCE CONDUCTING MATERNAL AND/OR PERINATAL DEATH AUDITS, THE TOTAL NUMBER OF FACILITIES IN THESE AREAS IS NOT A TRUE REPRESENTATION OF WHERE MPDSR MIGHT BE MOST USEFUL. WE EMPHASIZE IN THE TEXT THAT THE FACILITY SAMPLE IS IN NO WAY REPRESENTATIVE IN ANY COUNTRY AND INCLUDING ALL FACILITIES MIGHT IMPLY THAT WE WERE TRYING TO DO A NATIONAL SAMPLE AND COULD BE MISLEADING. ADDITIONALLY, PROVIDING THE TOTAL NUMBER OF FACILITIES WOULD BE A CHALLENGE AND COULD BE MISREPRESENTATIVE FOR THE FOLLOWING REASONS:

1) WE DO NOT KNOW THE DENOMINATOR OF FACILITIES CONDUCTING MPDSR PROCESSES IN EACH ADMINISTRATIVE UNIT FROM WHICH WE SAMPLED FACILITIES. 

2) DEFINING THE SMALLEST “ADMINISTRATIVE UNIT” CONSISTENTLY (E.G. SUB-DISTRICT, DISTRICT, REGION) COULD BE QUITE CHALLENGING GIVEN THE MIX OF HOSPITALS AND HEALTH CENTERS IN 3 OF THE 4 COUNTRIES AND THE INCLUSION OF HOSPITALS ONLY IN ZIMBABWE.

There is a mixture of health centres and hospitals in the sample. Why health centres? Do they all implement death reviews? Are health centres equipped to deal with severe morbidity? Are these patients not transferred to hospitals?

THANK YOU FOR RAISING THIS POINT. IN SOME COUNTRIES, HEALTH CENTRES CONDUCT DEATH REVIEWS, DEPENDING ON THEIR SIZE AND CAPACITY. IN THIS STUDY, ALL FACILITIES HAD TO MEET THE BASIC CRITERIA OF PROVISION OF CHILDBIRTH SERVICES, INCLUDING REFERRAL- AND PRIMARY-LEVEL FACILITIES, AND CURRENT OR RECENT EXPERIENCE CONDUCTING MATERNAL AND/OR PERINATAL DEATH AUDITS. HEALTH CENTRES IN THIS STUDY EXPERIENCED MATERNAL AND PERINATAL DEATHS, AND BREAKDOWNS IN REFERRAL SYSTEMS THAT COULD BE POTENTIALLY IDENTIFIED AND ADDRESSED THROUGH MPDSR.

WE DID LOOK AT THE DATA BETWEEN HEALTH CENTRES AND HOSPITALS. IN NIGERIA, ONLY THE FACILITIES WITH A HIGHER VOLUME OF REFERRAL CASES DEMONSTRATED EVIDENCE OF PRACTICE OR ABOVE. THE VOLUME OF FACILITY BIRTHS WAS NOT ASSOCIATED WITH A FACILITY’S MPDSR IMPLEMENTATION PROGRESS SCORE IN RWANDA AND ZIMBABWE. HIGHER-VOLUME REFERRAL FACILITIES IN TANZANIA SCORED HIGHER THAN HEALTH CENTRES IN GENERAL, BUT FEW HOSPITALS WERE FOLLOWING THE NATIONAL GUIDELINE COMPLETELY, INCLUDING WITH RESPECT TO INFORMATION FLOW TO OTHER LEVELS AND LITTLE COMMUNITY FOLLOW-UP. FURTHER RESEARCH IS NEEDED TO EXAMINE THE VALUE OF DEATH REVIEWS AT HEALTH CENTRES, WHICH WAS BEYOND THE SCOPE OF THIS STUDY. WE HAD ADDED THIS POINT IN THE LIMITATIONS SECTION. 

In Rwanda, no stakeholder was interviewed. Why? Is it really only a question of availability?

ALL OF THE NATIONAL AND SUB-NATIONAL STAKEHOLDERS WERE ENGAGED IN A MEETING THE WEEK OF THE ASSESSMENT AND NOT AVAILABLE FOR INTERVIEWS. WE HAVE ADDED THIS TO THE LIMITATION SECTION. 

Table 3 (and Figure 3) is quite interesting since the authors described the rationale or their hypotheses on which the markers are based, and allows some discussion. Did they try a sensitivity analysis to estimate the effect of modifying a score?

WE HAVE ADDED IN TEXT AROUND THE TOOL DEVELOPMENT IN THE METHODS SECTION AS WELL AS INCLUDED LEARNINGS ABOUT THE USE OF THE TOOL IN THE DISCUSSION SECTION. AS PART OF THE PROCESS OF DEVELOPMENT, WE ASSESSED THE FACE VALIDITY BY GROUNDING THE CONSTRUCTS IN THE LITERATURE ON THE TOPIC, ENGAGING EXPERTS IN THE DEVELOPMENT OF THE CRITERIA AND CONSULTING GLOBAL AND NATIONAL GUIDELINES. ASSESSING OTHER FORMS OF VALIDITY WERE BEYOND OUR SCOPE. THIS TOOL SOUGHT TO CLASSIFY PROGRESS MARKERS OF MPDSR PROCESSES. ITS SENSITIVITY IN BEING ABLE TO CORRECTLY IDENTIFY A FACILITY’S ABILITY TO DEMONSTRATE SPECIFIC IMPLEMENTATION MARKERS WAS NOT FORMALLY ASSESSED IN COMPARISON TO ALTERNATIVE ASSESSMENT TOOLS. DIFFERENCES IN THE APPLICATION OF THE TOOL BY VARIED TYPOLOGIES OF USERS IS NOTED IN THE LIMITATION SECTION, AS A CONCERN IN FUTURE USE. 

Line 140: how many assessors for the same facility? Did the authors find any discrepancies between the data collectors?

WE HAVE ADDED TO THE TEXT THE SIZE OF THE ASSESSMENT TEAMS FOR EACH FACILITY, WHICH VARIED FROM 2-5 PEOPLE. THE DATA COLLECTION TOOL ITSELF INCLUDES A SECTION FOR DATA COLLECTORS TO REFLECT ON FINDINGS, AS SHOWN IN THE APPENDIX. THE DATA COLLECTORS USED THIS REFLECTION PROCESS TO GUIDE THE DISCUSSION AROUND ANY DISCREPANCIES AND COME TO CONSENSUS ON THE FINDINGS. THERE IS A STATEMENT IN THE LIMITATIONS REGARDING POSSIBLE VARIABILITY ACROSS ASSESSMENT TEAMS.

Line 146: the fact that, facilities with a score of less than 10 points were excluded clearly means that only those facilities with some success in implementation are investigated. What is the reason explaining why these facilities did not reach the minimum score of 10 points? 

THANK YOU FOR RAISING THIS POINT; YOUR UNDERSTANDING OF WHY THESE FACILITIES WERE EXCLUDED IS CORRECT. THE ORIGINAL SAMPLE OF 55 WAS REDUCED TO INCLUDE FACILITIES AT THE IMPLEMENTATION PHASE FROM WHICH LESSONS COULD BE LEARNED ABOUT THE ACTUAL PRACTICE OF MPDSR IN FACILITIES. THE REASONS EXPLAINING WHY THESE FACILITIES DID NOT ATTAIN THE MINIMUM IMPLEMENTATION SCORE OF 10 POINTS WAS NOT INVESTIGATED BEYOND WHAT WE ASSESSED FOR IMPLEMENTATION FACTORS. 

What is then the meaning of an average score (very precise!) calculated only for the best implementers (line 173)?

THE AVERAGE SCORE DEMONSTRATES THE AVERAGE LEVEL OF IMPLEMENTATION ACROSS ALL THE FACILITIES DEMONSTRATING EVIDENCE OF PRACTICE AT THE TIME OF THE ASSESSMENT (I.E. THOSE THAT SCORED MORE THAN 10 POINTS) INCLUDED IN THE FINAL SAMPLE AND THEN ASSESSED FOR IMPLEMENTATION FACTORS.

OUR STUDY FACILITY SELECTION CRITERIA SPECIFIED THAT FACILITIES NEEDED TO MEET A MINIMUM STAGE OF IMPLEMENTATION (“EVIDENCE OF PRACTICE”), AND OUR ANALYSIS INCLUDED ONLY THESE FACILITIES. THUS THE AVERAGE SCORE MEASURES THE SPECIFIC STAGE OF IMPLEMENTATION, ON AVERAGE, ACROSS 3 PROGRESSIVE “IMPLEMENTATION STAGES” IN FACILITIES THAT DEMONSTRATED A MINIMUM STAGE OF IMPLEMENTATION. 

GIVEN OUR STUDY’S FOCUS ON ASSESSING FACTORS RELATED TO IMPLEMENTATION OF MPDSR PROCESSES (AS CONTRASTED TO “PRE-IMPLEMENTATION” READINESS FACTORS) WE DID NOT CONDUCT ANY ANALYSIS OF FACILITIES THAT DID NOT MEET A MINIMUM STAGE OF IMPLEMENTATION AS SPECIFIED IN OUR SELECTION CRITERIA (“EVIDENCE OF PRACTICE”).

I wonder why in the progress markers there was nothing about the coverage of deaths reviewed, i.e. the number of maternal deaths reviewed on the total of maternal deaths during the period and the same for perinatal deaths. This may be a well standardized indicator useful for comparing facilities if the policy is to review each case, of course. A second indicator would be the capacity of a facility to stop/prevent the problems/mismanagement that contributed to the deaths. THANK YOU FOR NOTING THESE IMPORTANT POINTS. ONE OF THE REASONS THAT COVERAGE WAS NOT INCLUDED AS AN INDICATOR IS BECAUSE NOT ALL SETTINGS HAD A MANDATE TO REVIEW 100% OF DEATHS THROUGH THE MPDSR PROCESS. WE AGREE THAT FURTHER WORK NEEDS TO BE DONE TO STRENGTHEN THIS MEASUREMENT PROCESS INCLUDING CONSIDERATION AROUND COVERAGE AS WELL AS THE FACILITY’S ABILITY TO MANAGE PROBLEMS THAT CONTRIBUTE TO DEATHS. WE HAVE ADDED A SECTION TO THE DISCUSSION FOCUSING ON THE LEARNINGS FROM THE DEVELOPMENT AND TESTING OF THE IMPLEMENTATION TOOL. 

Results

Table 4 is not easy to understand. Actually, each stage seems independent from the others while in reality it is difficult to understand that a facility without meeting minutes available, action items and follow-up have regular meetings and engagement.

THANK YOU FOR FLAGGING THIS ISSUE. WE FULLY AGREE THAT THESE STAGES REPRESENT A LINKED PROGRESSION IN IMPLEMENTATION AND SHOULD NOT BE CONSIDERED INDEPENDENT FROM EACH OTHER. FOR THIS REASON, WE REPORT IMPLEMENTATION SCORES BASED ON INDIVIDUAL PROGRESS MARKERS FOR EACH STAGE (RATHER THAN STAGE-SPECIFIC IMPLEMENTATION SCORES). THE PURPOSE OF TABLE 4 IS TO ALLOW READERS TO VISUALIZE ALL PROGRESS MARKERS IN A SINGLE TABLE AND TO BE ABLE TO APPRECIATE THAT INDIVIDUAL STAGES ARE NOT FULLY INDEPENDENT OF ONE ANOTHER.

To come back on the suggestion of building an indicator that identifies the progress in correcting mismanagement, the verbatim line 227 (‘Providing information about preventable factors that contribute to maternal death and using information to guide actions is key for preventing similar death in the future.’ –Facility interview, Rwanda’) shows that this is also a demand from the health workers and something measurable.

THANK YOU FOR NOTING THIS, WE FULLY AGREE. WE HAVE ADDED A SECTION TO THE DISCUSSION SECTION ABOUT THE PROCESS OF DEVELOPING THE TOOL, LESSONS LEARNED AND ADDITIONAL MARKERS TO CONSIDER.

It is surprising to read that line 232: Few facilities had agreements or procedures in place regarding the conduct of MPDSR meetings (9%). All teams were supposed to have been trained. This should be discussed.

THANK YOU FOR FLAGGING THIS IMPORTANT POINT. WE HAVE ADDED THIS TO THE DISCUSSION WITH REFERENCES. 

Line 298: “Though national guidelines included schematics on the reporting structure, including how responses should be tracked, less than one-third (28%) of facilities reported a formal written documentation system for tracking follow-up of recommended actions. Only one facility each in Zimbabwe and Tanzania and three in Rwanda demonstrated a formal process for follow-up of recommendations, apart from reviewing minutes at the next mortality audit meeting. None of the facilities in Nigeria had a systematic process for following up on recommendations.” This seems to be an interesting synthesis indicator of the effectiveness of maternal and perinatal death reviews.

THANK YOU FOR NOTING THIS. WE HAVE ADDED A SECTION TO THE DISCUSSION SECTION ABOUT THE ADAPTED TOOL, LESSONS LEARNED AND KEY INDICATORS TO CONSIDER. 

I appreciated also the ST4 with a trial to build a full synthesis indicator that considers each successive step. The progress marker ‘There is evidence of change based on recommendations that arise from death review findings’ reached by 45% of facilities is an achievement, even in a sample of facilities supported by the USAID program.

THANK YOU FOR RECOGNIZING THIS AS AN ACHIEVEMENT. WE ALSO AGREE.

The section of enablers and barriers is really small and does not add to what we already know.

THANK YOU FOR NOTING THIS ISSUE. WE HAVE EXPANDED THIS SECTION WITH ADDITIONAL FINDINGS GIVEN ONE OF OUR MAIN OBJECTIVES WAS TO UNDERSTAND THE ENABLERS AND BARRIERS OF IMPLEMENTATION. MORE DETAILS ARE ADDED IN THE SUPPLEMENTARY FILES AROUND THESE FINDINGS. 

Discussion

The discussion section does not discuss the limitations of the method itself. To what extent the scoring sufficiently reflects the effectiveness of the implementation of these death reviews?

 THANK YOU FOR NOTING THIS CONCERN. WE HAVE ADDED SUBSTANTIAL ADDITIONAL TEXT TO THE LIMITATION SECTION AS WELL AS SECTION ABOUT THE LEARNINGS AROUND THE DEVELOPMENT AND USE OF THE IMPLEMENTATION SCORING TOOL.

The discussion section is supported by few references (8x #22, 7x #23, 4x #32 and 9 other references), see below ‘supporting literature’.

Supporting literature

Most of the cited literature are documents or articles from WHO or WHO civil servants, then from US authors. Table 3, explaining the rationale for the items in the score, is mainly based on 3 references (#21 13x; #22 5x; and #23 5x) while the discussion is based mainly on 2 references (#22 8x and #23 7x).

However, important literature on clinical audits and maternal death reviews comes from different groups of researchers. Among them: Johnston G, Crombie I K, Alder E M, Davies H T O and Millard A. Reviewing audit: barriers and facilitating factors for effective clinical audit Qual. Health Care 2000;9;23-36. doi:10.1136/qhc.9.1.23; Ivers N, Jamtvedt G, Flottorp S, Young JM, Odgaard-Jensen J, French SD, O’Brien MA, Johansen M, Grimshaw J, Oxman AD. Audit and feedback: effects on professional practice and healthcare outcomes. Cochrane Database of Systematic Reviews 2012, Issue 6. Art. No.: CD000259. DOI: 10.1002/14651858.CD000259.pub3; Filippi V, Brugha R, Browne E, Gohou V, Bacci A, De Brouwere V, Sahel S, Goufodji S, Alihonou E, Ronsmans C. 2004. How to do (or not to do) . . . Obstetric audit in resource poor settings: lessons from a multi-country project auditing ‘near miss’ obstetrical emergencies. Health Policy and Planning, 19(1), 57-66; Müffler N, Trabelssi M, De Brouwere V. 2007. Scaling up clinical audits of obstetric cases in Morocco. Tropical Medicine & International Health 12(10), 1248-1257; and the numerous papers from Kongnyuy et al. exploring maternal death reviews in Malawi).

Please, note that the references are not standardized: sometimes, the reference is with the acronym of the journal (bjog), sometimes in full (Health policy & planning) and careful attention should be paid to have a correct list.

 THANK YOU FOR PROVIDING ADDITIONAL LITERATURE FOR US TO REVIEW AND INCLUDE. WE ALSO APPRECIATE YOU FLAGGING ISSUES WITH THE REFERENCE FORMAT, WHICH WE HAVE CORRECTED. 

REVIEWER #2 

The rate of maternal and perinatal death in developing countries are still high, so it is urgent issue to prevent future deaths. Therefore, this study is very significant. A huge amount of data has been analyzed in detail and I think it is valuable information. Please tell me the following: THANK YOU. WE ARE PLEASED THAT YOU FOUND THE STUDY VERY SIGNIFICANT. 

1) It was written that you used a standardized scoring methodology. Which is the name of the score scale and which paper shows that the scale is standardized?

Is there a description of the validity of this measure in this paper?

 WE APPRECIATE YOUR QUESTION AND HAVE ADDED A NEW SECTION ON THE SCORING TOOL IN THE DISCUSSION TO PROVIDE MORE DETAILS ABOUT THE DEVELOPMENT OF THE TOOL AND WHAT WE LEARNED IN USING IT.

AS PART OF THE PROCESS OF DEVELOPMENT, WE ASSESSED THE FACE VALIDITY BY GROUNDING THE CONSTRUCTS IN THE LITERATURE ON THE TOPIC, ENGAGING EXPERTS IN THE DEVELOPMENT OF THE CRITERIA AND CONSULTING GLOBAL AND NATIONAL GUIDELINES. ASSESSING OTHER FORMS OF VALIDITY WERE BEYOND OUR SCOPE. THIS TOOL SOUGHT TO CLASSIFY PROGRESS MARKERS OF MPDSR PROCESSES. ITS SENSITIVITY IN BEING ABLE TO CORRECTLY IDENTIFY A FACILITY’S ABILITY TO DEMONSTRATE SPECIFIC IMPLEMENTATION MARKERS WAS NOT FORMALLY ASSESSED IN COMPARISON TO ALTERNATIVE ASSESSMENT TOOLS. DIFFERENCES IN THE APPLICATION OF THE TOOL BY VARIED TYPOLOGIES OF USERS IS NOTED IN THE LIMITATION SECTION, AS A CONCERN IN FUTURE USE. 

2) What is the mean implementation progress score in developed countries, especially those with low maternal and perinatal mortality?

 WE DO NOT KNOW THE MEAN IMPLEMENTATION PROGRESS SCORE IN ANY OTHER COUNTRIES OR CONTEXT SINCE THIS IS THE FIRST TIME THE TOOL USED IN THIS STUDY WAS APPLIED TO ASSESS MPDSR IMPLEMENTATION PROCESSES. FURTHER APPLICATION AND ADAPTATION OF THE TOOL IN ADDITIONAL SETTINGS COULD PROVIDE THIS INFORMATION.

3) It feels redundant overall, can you summarize it a little more concisely? You can leave the necessary information as it is.

 THANK YOU. WE HAVE ATTEMPTED TO REDUCE ANY REDUNDANCY AS WELL AS APPROPRIATELY ADDRESS COMMENTS FROM THE EDITOR AND THE REVIEWERS.

---

## [Decision Letter · Decision Letter 1]

10 Nov 2020

PONE-D-20-00503R1

“It might be a statistic to me, but every death matters.”: An assessment of facility-level maternal and perinatal death surveillance and response systems in four sub-Saharan African countries

PLOS ONE

Dear Dr. Kinney,

Thank you for submitting your manuscript to PLOS ONE. After careful consideration, we feel that it has merit but does not fully meet PLOS ONE’s publication criteria as it currently stands. Therefore, we invite you to submit a revised version of the manuscript that addresses the points raised during the review process.

We look forward to receiving your revised manuscript.

Kind regards,

Jennifer Yourkavitch

Academic Editor

PLOS ONE

Additional Editor Comments (if provided):

Thank you for responding to the reviewers' comments. It appears that you edited the reference list; however, there are no tracked changes. Please indicate those edits with tracked changes.

You noted that you based the tool on literature, expert opinion, and national policies; however, Table 3 indicates justifications for tool elements based only on literature. You mention policy differences at different points in the Discussion--around lines 410-420 and 475--but the reader doesn't get a sense of what the national policies contain, how they differ from each other, and how they differ from global standards in the main text. The information in Supplemental Files 1 and 2 is useful but we still can't see how the policies measure up against the tool. In other words, is the tool measuring things not contained in the policies for one or more countries? And might national policy deviance from literature or global standards account in part for the scoring? It would be useful to know if facility scores result more from a lack of adequate national policy or lack of capacity at facilities. You seem to imply the latter without addressing the former possibility. This issue would benefit from an organized discussion rather than mentioning policy issues in different places.

Reviewers' comments:

Reviewer's Responses to Questions

**Comments to the Author**

1. If the authors have adequately addressed your comments raised in a previous round of review and you feel that this manuscript is now acceptable for publication, you may indicate that here to bypass the “Comments to the Author” section, enter your conflict of interest statement in the “Confidential to Editor” section, and submit your "Accept" recommendation.

Reviewer #1: All comments have been addressed

Reviewer #2: All comments have been addressed

2. Is the manuscript technically sound, and do the data support the conclusions?

Reviewer #1: Yes

Reviewer #2: Partly

3. Has the statistical analysis been performed appropriately and rigorously? 

Reviewer #1: N/A

Reviewer #2: Yes

4. Have the authors made all data underlying the findings in their manuscript fully available?

Reviewer #1: Yes

Reviewer #2: Yes

5. Is the manuscript presented in an intelligible fashion and written in standard English?

Reviewer #1: Yes

Reviewer #2: Yes

6. Review Comments to the Author

Reviewer #1: just a few typos in the revised version, e.g. "characterstics" instead of "characteristics", line 543, p.28.

Also, please, standardize the references so that we don't have sometimes the full name of the journal and sometimes a short name.

Reviewer #2: (No Response)

7. PLOS authors have the option to publish the peer review history of their article (what does this mean?). If published, this will include your full peer review and any attached files.

Reviewer #1: **Yes: **Vincent De Brouwere

Reviewer #2: **Yes: **Mako Morikawa

---

## [Author Response · Author response to Decision Letter 1]

23 Nov 2020

EDITOR COMMENTS

It appears that you edited the reference list; however, there are no tracked changes. Please indicate those edits with tracked changes. 

THANK YOU FOR NOTING THAT THERE WERE NO TRACK CHANGES ON THE REFERENCE LIST. FOR THE REVISED MANUSCRIPT, WE USED A REFERENCING SOFTWARE TO UPDATE THE REFERENCES IN ORDER TO MINIMIZE HUMAN ERROR WHEN ADDRESSING THE CORRECTIONS. UNFORTUNATELY, THE SYSTEM DOES NOT ALLOW FOR TRACK CHANGES. FOR THIS REVISED RESUBMITTED VERSION, WE HAVE REMOVED THE SOFTWARE FROM THE MANUSCRIPT AND HAVE MADE THE PREVIOUS EDITS WITH TRACK CHANGE. 

You noted that you based the tool on literature, expert opinion, and national policies; however, Table 3 indicates justifications for tool elements based only on literature. 

THANK YOU FOR YOUR COMMENT. WE HAVE MADE EDITS TO CLARIFY THE POLICY MAPPING PROCESS AND HOW THIS LINKS TO THE RESULTS AND DISCUSSION. AS DESCRIBED, THE TOOL ITSELF WAS CONCEPTUALLY ADAPTED FROM A TOOL USED TO MEASURE KANGAROO MOTHER CARE IMPLEMENTATION. WE USED THE MPDSR LITERATURE, EXPERT OPINION AND THE EXISTING GLOBAL GUIDELINES FOR PERINATAL DEATH AUDIT AND MDSR TO INFORM THE CONTENT. WE HAD AMENDED THE SENTENCE TO REMOVE NATIONAL GUIDELINES AS A SOURCE TO INFORM THE TOOL CONTENT. REFERENCES TO THE GLOBAL GUIDELINES ARE INCLUDED AMONG THE REFERENCES IN TABLE 3. 

You mention policy differences at different points in the Discussion--around lines 410-420 and 475--but the reader doesn't get a sense of what the national policies contain, how they differ from each other, and how they differ from global standards in the main text. The information in Supplemental Files 1 and 2 is useful but we still can't see how the policies measure up against the tool. In other words, is the tool measuring things not contained in the policies for one or more countries? And might national policy deviance from literature or global standards account in part for the scoring? It would be useful to know if facility scores result more from a lack of adequate national policy or lack of capacity at facilities. You seem to imply the latter without addressing the former possibility. This issue would benefit from an organized discussion rather than mentioning policy issues in different places.

THE FINDINGS OF THE POLICY MAPPING ARE PRESENTED AT THE START OF THE RESULTS SECTION (LINES 172-180) WITH DETAILS IN SUPPLEMENTAL FILE 2 (TABLE S2.1). IT IS HELPFUL TO HAVE YOUR PERSPECTIVE REGARDING THE POSSIBLE CONFUSION AROUND THE LINKS OF THE TOOL AND THE POLICY ANALYSIS, AND WE APPRECIATE YOU FLAGGING THIS CONCERN. WE HAVE MADE EDITS TO THE DISCUSSION SECTION TO ADDRESS YOUR QUESTIONS.

PEER REVIEWER COMMENTS

REVIEWER #1 

just a few typos in the revised version, e.g. "characterstics" instead of "characteristics", line 543, p.28.

THANK YOU. WE HAVE MADE THIS CORRECTION AND HAVE DONE ANOTHER REVIEW TO CHECK FOR OTHER TYPOS.

Also, please, standardize the references so that we don't have sometimes the full name of the journal and sometimes a short name. 

THANK YOU FOR NOTING THE ISSUES IN THE REFERENCES. THE JOURNAL’S REFERENCING STYLE, VANCOUVER, REQUIRES THE TITLES OF JOURNALS TO BE ABBREVIATED AS THEY APPEAR IN THE JOURNALS IN NCBI DATABASES. HTTPS://WWW.NCBI.NLM.NIH.GOV/NLMCATALOG/JOURNALS WE HAVE GONE THROUGH EACH REFERENCE AND CORRECTED ACCORDINGLY. 

REVIEWER #2 

No comments

THANK YOU.

---

## [Editor Report · Decision Letter 2]

30 Nov 2020

“It might be a statistic to me, but every death matters.”: An assessment of facility-level maternal and perinatal death surveillance and response systems in four sub-Saharan African countries

PONE-D-20-00503R2

Dear Dr. Kinney,

We’re pleased to inform you that your manuscript has been judged scientifically suitable for publication and will be formally accepted for publication once it meets all outstanding technical requirements.

Kind regards,

Jennifer Yourkavitch

Academic Editor

PLOS ONE
---

## [Editor Report · Acceptance letter]

7 Dec 2020

PONE-D-20-00503R2 

*“It might be a statistic to me, but every death matters.”:* An assessment of facility-level maternal and perinatal death surveillance and response systems in four sub-Saharan African countries 

Dear Dr. Kinney:

I'm pleased to inform you that your manuscript has been deemed suitable for publication in PLOS ONE. Congratulations! Your manuscript is now with our production department. 

Kind regards, 

on behalf of

Dr. Jennifer Yourkavitch 

Academic Editor

PLOS ONE